# Advanced Cutaneous Squamous Cell Carcinoma: Biology, Immunotherapy, and Evolving Prognostic Factors

**DOI:** 10.3390/biomedicines13123010

**Published:** 2025-12-08

**Authors:** Antonio Di Guardo, Federica Trovato, Carmen Cantisani, Francesco Ricci, Giovanni Di Lella, Francesco Moro, Roberto Morese, Annarita Panebianco, Steven P. Nisticò, Giovanni Pellacani, Luca Fania

**Affiliations:** 1Unit of Dermatology, Department of Medical and Cardiovascular Sciences, “Sapienza” University of Rome, 00161 Rome, Italy; federica.trovato@uniroma1.it (F.T.); carmen.cantisani@uniroma1.it (C.C.); f.moro@idi.it (F.M.); steven.nistico@uniroma1.it (S.P.N.); giovanni.pellacani@uniroma1.it (G.P.); 2IRCCS Istituto Dermopatico dell’Immacolata (IDI-IRCCS), Dermatological Research Hospital, 00167 Rome, Italy; f.ricci@idi.it (F.R.); g.dilella@idi.it (G.D.L.); r.morese@idi.it (R.M.); a.panebianco@idi.it (A.P.); l.fania@idi.it (L.F.); 3Department of Life Science, Health, and Health Professions, Link University of Rome, 00165 Rome, Italy

**Keywords:** cutaneous squamous cell carcinoma, cemiplimab, PD-1 blockade, immunotherapy, prognostic factors, special populations

## Abstract

Advanced cutaneous squamous cell carcinoma (aCSCC) comprises locally advanced and metastatic disease not amenable to curative surgery or radiotherapy and is associated with substantial morbidity, mortality, and healthcare costs. This narrative review summarizes current knowledge on the epidemiology, biology, clinical presentation, and staging of aCSCC and critically appraises therapeutic strategies with a focus on programmed death 1 (PD-1) blockade. Immune checkpoint inhibitors now represent the main systemic treatment for advanced cSCC, with clinical trials and observational studies reporting response rates around 45–60%, sustained benefit in a subset of patients, and a manageable yet clinically relevant profile of immune-related toxicities. However, outcomes remain heterogeneous, particularly in elderly, comorbid, and immunosuppressed patients. We therefore review established and emerging prognostic determinants spanning clinical, anatomical, histopathological, metabolic, inflammatory, and on-treatment domains. Priorities for biomarker-enriched studies and harmonized real-world registries to enable more refined risk stratification and genuinely personalized, multidisciplinary management of aCSCC are also outlined.

## 1. Introduction

Cutaneous squamous cell carcinoma (cSCC) is one of the most prevalent malignancies affecting the skin, ranking second only to basal cell carcinoma among nonmelanoma skin cancers [1,2]. Its global incidence has been steadily increasing over recent decades, with an estimated annual rise of up to 50–200%, making it a significant and growing public health concern [2,3]. While the majority of cSCCs are amenable to surgical excision and have a favorable prognosis, with five-year survival rates exceeding 90%, a subset of tumors exhibits aggressive behaviour, marked by a higher likelihood of local invasion, recurrence, and metastasis [4,5]. Though metastasis is relatively rare compared to other malignancies, it remains clinically relevant due to its association with poor outcomes and its potential to affect lymph nodes and distant organs. The clinical spectrum of cSCC varies widely, ranging from small, well-differentiated, indolent tumors to rapidly growing, poorly differentiated lesions capable of deep tissue destruction involving cartilage or bone [6]. In detail, advanced cutaneous squamous cell carcinoma (aCSCC) comprises locally advanced (laCSCC) and metastatic cutaneous squamous cell carcinoma (mCSCC) not amenable to curative surgery or curative radiotherapy, or both [7]. Risk stratification is therefore essential and must account for clinical presentation, anatomical site, histological features, and patient-related factors. In parallel with the rising incidence of cSCC, the field of dermatology has witnessed significant technological and therapeutic advances aimed at improving patient care. Diagnostic capabilities have been enhanced by the increasing integration of non-invasive imaging modalities, such as dermoscopy, reflectance confocal microscopy (RCM), and line-field confocal optical coherence tomography (LC-OCT), which allow for real-time, in vivo assessment of suspicious lesions and help refine clinical decision-making [8,9]. On the therapeutic front, the treatment paradigm for cSCC has expanded beyond conventional surgery and radiotherapy, particularly for cases that are locally advanced, recurrent, or metastatic. The introduction of immune checkpoint inhibitors has marked a transformative shift in the management of advanced cSCC, offering durable responses in a subset of patients with limited therapeutic options. Among these, cemiplimab, an anti-PD-1 monoclonal antibody, has become the first systemic immunotherapy approved for patients with unresectable or metastatic cSCC, demonstrating favorable efficacy and safety profiles in both clinical trials and real-world settings [10,11].

This narrative review aims to provide a practice-oriented synthesis of advanced cutaneous squamous cell carcinoma, from epidemiology, biology, and staging to current multimodal treatment, with particular focus on PD-1 blockade and cemiplimab. We summarize and contextualize evidence from clinical trials and real-world cohorts on effectiveness and safety, including immune-related adverse events, within contemporary multidisciplinary care. At the same time, we acknowledge the persisting unmet needs and knowledge gaps, such as incomplete prognostic stratification, limited data in vulnerable subgroups, and heterogeneous real-world outcomes, of which only some can be addressed given the current literature. In parallel, we discuss available and emerging prognostic factors across clinical, pathological, metabolic, inflammatory, and on-treatment domains, with specific attention to special populations, and highlight key priorities for future biomarker-enriched studies and harmonized registries to support more personalized, risk-adapted management.

## 2. Materials and Methods

A comprehensive literature search was conducted independently in each database (PubMed/MEDLINE and Scopus) to identify publications relevant to aCSCC, with complementary searches in Embase, Web of Science, and the Cochrane Library to enhance coverage. Trial registries (ClinicalTrials.gov and World Health Organization—International Clinical Trials Registry Platform) and major conference abstract repositories (American Society of Clinical Oncology, European Society for Medical Oncology, American Academy of Dermatology) were screened for recent and ongoing studies not yet indexed in bibliographic databases. No language or publication date restrictions were applied. The search covered database inception through November 2025. The search strategy combined controlled vocabulary (e.g., MeSH/Emtree) and free-text terms, adapted to each database, using Boolean operators and truncation where appropriate. The core query paired disease terms with treatment, outcomes, and prognostic constructs. Representative terms included, in various combinations: “cutaneous squamous cell carcinoma” OR “cSCC” OR “skin squamous cell carcinoma” AND “advanced” OR “locally advanced” OR “unresectable” OR “metastatic” AND “immunotherapy” OR “immune checkpoint inhibitor” OR “anti–PD-1” OR “cemiplimab” OR “pembrolizumab” OR “nivolumab” OR “checkpoint blockade” OR “EGFR inhibitor” OR “cetuximab” OR “chemotherapy” OR “platinum” OR “radiotherapy” OR “neoadjuvant” OR “adjuvant” OR “multimodal” AND “real-world” OR “observational” OR “registry” OR “cohort” OR “retrospective” OR “prospective” AND “prognostic” OR “risk factor” OR “staging” OR “AJCC” OR “Brigham and Women’s (BWH)” OR “perineural invasion” OR “tumor thickness” OR “depth of invasion” OR “immunosuppressed” OR “solid organ transplant” AND “overall survival” OR “progression-free survival” OR “objective response rate” OR “duration of response” OR “disease control” OR “toxicity” OR “immune-related adverse events” OR “quality of life”. Reference lists of all eligible articles and relevant reviews were manually screened to identify additional studies. Regulatory assessment reports (e.g., European Medicines Agency/Food and Drug Administration labels and reviews) were consulted to contextualize indications and pivotal evidence. Given the narrative scope of this review, no formal meta-analysis was planned. Nevertheless, we applied a structured approach to study selection and appraisal. After duplicate removal, titles and abstracts were screened for relevance, followed by full-text assessment. Preference was given to peer-reviewed original research, including randomized and non-randomized clinical trials, prospective and retrospective cohorts, registries, and large case series reporting predefined clinical outcomes. Studies focusing on non-cutaneous squamous cell carcinomas (e.g., mucosal head-and-neck, anogenital) were excluded. Single-patient case reports were generally excluded unless they provided unique insights pertinent to advanced disease management. The study selection process is summarized in the PRISMA flow diagram (Figure 1).

To guide selection in a reproducible manner while preserving the breadth expected of a narrative synthesis, we used a PICO framework: (i) Population: Adults with advanced cSCC, defined as unresectable locally advanced or metastatic disease; special populations (immunosuppressed, solid-organ transplant recipients) were included when reported; (ii) Intervention/Exposure: Systemic therapies (anti–PD-1 agents such as cemiplimab, pembrolizumab; EGFR inhibitors; cytotoxic chemotherapy), and multimodal strategies integrating radiotherapy and/or surgery; exposures also included prognostic factors and staging systems relevant to advanced disease; (iii) Comparator: Alternative systemic regimens, best supportive care, historical or real-world comparators, or no comparator in single-arm studies; (iv) Outcomes: Treatment effectiveness (ORR, CR, DOR, DCR, PFS, OS), safety (overall and grade ≥ 3 AEs, immune-related AEs), patient-reported outcomes, and prognostic performance (effect sizes, discrimination/calibration).

## 3. Results

Advanced cutaneous squamous cell carcinoma (aCSCC) is broadly classified into locally advanced (laCSCC) and metastatic (mCSCC) disease, the latter encompassing both locoregional and distant metastatic spread [12]. According to the current European guidelines, laCSCC is defined as non-metastatic cSCC that is not amenable to curative surgery or radiotherapy due to factors such as multiple recurrences, extensive tumour size, bone erosion, deep invasion beyond the subcutaneous tissue into muscle or along major nerves, or anatomical locations where curative resection would result in unacceptable functional impairment, morbidity, or disfigurement. Conversely, mCSCC refers to tumours with regional lymph node involvement and/or distant metastases.

### 3.1. Epidemiology

cSCC represents the second most frequent cutaneous malignancy after basal cell carcinoma (BCC), with a rapidly increasing incidence worldwide [1]. Epidemiological data for advanced cSCC are limited due to inconsistent reporting and the lack of a uniform definition [12]. The frequency of metastasis in unselected cSCC cohorts is estimated at 3–5% for nodal spread and <1% for distant metastasis [13]. A large retrospective study in the United States reported local recurrence in 4.6% of patients, nodal metastasis in 3.7%, distant metastasis in 0.4%, and disease-specific death in 2.1% [14]. A UK study of over 76,000 patients (National Cancer and Analysis Registration Service—NCRAS, 2013–2015) documented a cumulative incidence of metastatic cSCC of 2.1%, with rates of 2.4% in men and 1.1% in women [15]. Data from Germany (1997–2011) indicated that 2.2% of cases were regional (direct invasion or lymph node involvement) and 0.3% were distant metastases, while in Norway (1963–2011) 2.1% of cSCCs were classified as advanced [16,17]. Importantly, the majority of metastatic events occur early: 85% within the first 2 years and over 90% within 3 years of the primary tumor diagnosis [15]. Survival outcomes are significantly worse for advanced cSCC. In the UK NCRAS dataset, the 3-year survival for patients with mcSCC was 46% in men and 29% in women, compared with 65% and 68% for non-metastatic cases [15]. German cancer registry data reported a relative 5-year survival of 94% for all cSCC cases, but only 58.3% for those with regional metastasis [16]. Norwegian data showed 5-year relative survival rates of 82% in men and 88% in women with localized cSCC, dropping to 51% and 64%, respectively, in advanced disease [17]. The proportion of advanced cSCC among selected populations is illustrated in Figure 2. Distant metastases confer the worst prognosis, with studies showing an eightfold increased risk of death compared to nodal metastasis; 89% of patients with distant spread die within 5 years [18]. Immunosuppressed patients, particularly solid organ transplant recipients (SOTRs), represent a high-risk group for advanced cSCC. The incidence of cSCC in SOTRs is 65–100 times higher than in the general population [19,20]. A meta-analysis reported a pooled metastatic risk of 7.3% in SOTRs compared with 3.1% in immunocompetent patients [21]. In a retrospective cohort of 51 SOTRs with advanced cSCC, 88% had metastatic disease, and the 5-year disease-specific survival was only 30.5% [22]. Aggressive disease tends to present at a younger age in this group, with median diagnosis around 62 years. The time between primary tumor detection and metastasis is often short, averaging 1.4 years [22,23].

### 3.2. Pathogenesis and Evolution to Advanced cSCC

#### 3.2.1. Initiating Drivers and Early Carcinogenesis

cSCC arises through the convergence of environmental injury, impaired immune surveillance, and inherited or acquired genomic alterations that together drive a multistep carcinogenic process from actinic keratosis to invasive disease (Figure 3) [24,25,26,27].

Chronic ultraviolet radiation (UVR) is the dominant initiator: UVB and UVA generate cyclobutane pyrimidine dimers and 6–4 photoproducts with characteristic C>T transitions at dipyrimidine sites, seeding early driver mutations, most notably in TP53, that permit clonal survival under continued UV stress [25,28]. Risk scales with cumulative exposure (outdoor work, high-insolation latitudes), distinguishing cSCC from BCC’s stronger link to intermittent exposure and sunburns [28,29,30]. Artificial or therapeutic sources add to this burden: indoor tanning at young ages, PUVA in psoriasis (dose-dependent), and ionizing radiation confer substantial additional risk [31]. Chemical carcinogens (arsenic, polycyclic aromatic hydrocarbons, nitrosamines) further damage DNA via adduct formation and hinder repair [32,33]. Immunosuppression is a powerful amplifier: solid organ transplant recipients experience 65–250-fold higher incidence, reflecting reduced immune surveillance, UV synergy, and drug-specific carcinogenicity (e.g., azathioprine, cyclosporine) [34,35]. Elevated risks also occur in hematologic malignancies (CLL, NHL) and advanced HIV infection, where anal SCC is notably enriched, owing to combined cellular and humoral immune defects [36,37]. Beta-HPVs (types 5, 8, 38) likely act as co-carcinogens in sun-exposed skin by E6/E7-mediated interference with p53 and Rb, especially early in tumorigenesis and in immunosuppressed or genetically predisposed hosts (e.g., epidermodysplasia verruciformis) [38,39]. Chronic inflammation, scarring, and non-healing wounds create a pro-tumor microenvironment rich in reactive oxygen species and cytokines (Marjolin ulcers) [40]. Dysregulated TGF-β signaling and an imbalanced IL-18/IL-37 axis further supports immune evasion and aggressive behavior, including in oral SCC [41,42]. Certain drugs modulate risk: BRAF inhibitors can trigger eruptive keratoacanthomas/cSCC via paradoxical MAPK activation in RAS-mutant keratinocytes, an effect mitigated by MEK co-inhibition, while long-term voriconazole and vismodegib have also been linked to increased cSCC incidence [43,44]. Lifestyle factors modestly contribute (smoking, alcohol) and may interact with exposure patterns from occupational vs. recreational activity [45]. Inherited cancer-prone states, such as nucleotide excision repair defects (xeroderma pigmentosum), pigmentation disorders (oculocutaneous albinism), viral susceptibility syndromes (EV), and chronic-wounding genodermatoses (recessive dystrophic epidermolysis bullosa), underscore the primacy of DNA repair, photoprotection, and cutaneous barrier integrity [46,47,48]. Beyond rare syndromes, GWAS highlights common, low-to-moderate effect alleles in pigmentation (*MC1R*, *TYR*, *IRF4*, *SLC45A2*, *HERC2*), immunity (*HLA-DQA1*, *FOXP1*), and keratinocyte regulation (*TP63*, *BNC2*, *DEF8*) that shape baseline susceptibility [49,50]. Molecularly, early *TP53* inactivation, *CDKN2A* silencing, and frequent truncating mutations in *NOTCH1/2* disrupt apoptosis and differentiation; context-dependent RAS activation (notably with BRAF inhibition), *KNSTRN* mutations, *MYC* amplification, copy-number losses (3p, 9p; *FHIT*, *PTPRD*), and epigenetic dysregulation collectively propel progression within a UV-damaged “field” of mutated clones [51,52,53]. Tumor–stroma crosstalk (angiogenic signaling such as VEGF), recruitment of immunosuppressive cells, MHC-I downregulation, immunosuppressive cytokines (IL-10, TGF-β), and PD-L1 expression consolidate immune escape [54,55]. The main pathogenetic underlying advanced cutaneous squamous cell carcinoma are summarized in Table 1.

#### 3.2.2. Invasion and Dissemination Mechanisms

The progression from cSCC in situ to an advanced and metastatic phenotype is a consequence of stepwise genetic and epigenetic alterations that drive changes in tumor cell behavior and interaction with the surrounding microenvironment. In the transition from in situ to invasive disease, the loss of cell–cell adhesion molecules, such as desmocollin-3 (*DSC3*), desmoplakin (*DSP*), and junction plakoglobin (*JUP*), plays a central role in enabling keratinocytes to breach the basement membrane [56]. This functional impairment, which correlates with somatic mutations and reduced protein expression, facilitates dermal infiltration by disrupting desmosomal integrity and intercellular cohesion. As tumor cells migrate deeper into the dermis, they acquire additional alterations, including mutations in genes such as *PTEN*, *NOTCH1*, and *CDKN2A*, that contribute to proliferative advantage, immune evasion, and matrix degradation [57]. Notably, whole-exome sequencing has revealed a distinct mutational signature in the deeper invasive front, enriched in pathways regulating cell junction organization and epithelial–mesenchymal transition (EMT), suggesting a spatial evolution of tumor clones under selective pressure [58]. Local invasion sets the stage for perineural and lymphovascular involvement. Perineural spread, associated with mutations in adhesion and axon guidance genes such as *NTNG2* and *RELN*, enables tumor extension along nerve sheaths, often without overt mass formation [56,59]. Simultaneously, entry into the lymphatic system is facilitated by tumor-induced lymphangiogenesis and upregulation of VEGF-C/D, promoting access to regional lymph nodes [60]. Clonal selection of subpopulations with increased motility and immune-modulatory capabilities, such as PD-L1 overexpression, further supports immune escape and dissemination [61]. *CD274* (gene encoding PD-L1) point mutations are uncommon and are not considered major driver events. Ultimately, hematogenous metastasis occurs in the context of profound genomic instability. Copy number alterations, gain-of-function mutations in driver oncogenes (e.g., *EGFR*), and loss of tumor suppressors like *TP53* and *FAT1* contribute to the acquisition of traits necessary for distant colonization [62,63]. These include survival in circulation, endothelial adhesion, and growth in foreign microenvironments. Thus, advanced cSCC represents the culmination of a dynamic evolutionary process shaped by both intrinsic tumor biology and host–tumor interactions.

### 3.3. Clinical Features, Diagnostic Work-Up, Histology, and Staging

Clinically, aCSCC may present as an enlarging, indurated plaque or nodule with hyperkeratosis, ulceration, or exophytic growth and often ill-defined margins, with rapid enlargement, friability, hemorrhagic crusting, and pain more common in poorly differentiated lesions [1,6]. High-risk anatomical sites (non-glabrous lip, ear, periocular and perinasal regions) are prone to aggressive local behavior and recurrence owing to complex anatomy [64]. Tumors arising in scars or chronic wounds (Marjolin ulcers) show heightened risk of nodal involvement and, in some series, synchronous distant spread [65]. Squamous cell carcinoma of the lip, frequently on a background of actinic cheilitis, behaves more aggressively with a greater likelihood of perineural invasion and regional spread [66]. Perineural invasion (PNI) is an adverse feature: neuropathic pain, paresthesia, hypoesthesia, or weakness along a named nerve distribution should raise suspicion and prompt targeted imaging and multidisciplinary planning [67]. When metastatic, patients may present with firm, sometimes fixed lymphadenopathy (e.g., parotid/cervical for head and neck primaries; axillary/inguinal for extremity primaries) or with organ-specific symptoms. Lung is the most frequent distant site, followed by bone and other viscera [12]. Host immunosuppression further tilts the course toward advanced disease [64]. aCSCC accounts for disproportionate morbidity, healthcare utilization, and mortality relative to overall cSCC burden [12,67,68]. Major prognostic determinants for metastatic spread in invasive cSCC are summarized in Table 2.

#### 3.3.1. Histopathology

Advanced tumors span the differentiation spectrum: well-differentiated aCSCC shows infiltrative nests and lobules of atypical squamous cells with intercellular bridges and keratin pearls; progressive invasion manifests as irregular tongues extending into deep dermis, subcutis, and sometimes muscle or bone. Poorly differentiated tumors exhibit reduced keratinization, marked pleomorphism, brisk mitotic activity, and necrosis. immunohistochemistry may be required to exclude mimics [69,70]. Adverse variants, such as desmoplastic, acantholytic, clear cell, pseudovascular, mucinous, pigmented, and tumors arising in scars or chronic ulcers, are enriched among advanced presentations; desmoplastic SCC, in particular, carries a high local recurrence risk and frequent PNI [71]. Keratoacanthoma-like lesions with pronounced atypia are managed as SCC given potential for invasion/metastasis [71]. Pathology reports in aCSCC should quantify tumor thickness (with >6 mm especially adverse), level of invasion (beyond subcutaneous fat), margin status, degree of differentiation, lymphovascular invasion, and granular PNI metrics (nerve caliber ≥ 0.1 mm, involvement of named nerves, proximal extent), as these parameters drive staging, adjuvant therapy, and follow-up [69,70,71].

#### 3.3.2. Staging and Risk Stratification

Staging underpins prognosis and treatment selection. For cSCC of the head and neck, AJCC 8th edition TNM is used; for sites outside head and neck, AJCC 7th remains a common reference for clinical staging (institution-dependent) [72,73]. Because conventional TNM incompletely captures metastatic risk for cutaneous primaries, the Brigham and Women’s Hospital (BWH) system complements staging by tallying high-risk features—e.g., diameter ≥ 2 cm, poor/undifferentiated histology, PNI (including caliber-based criteria or named nerves), deep invasion (>6 mm or beyond fat), high-risk location (ear or non-glabrous lip), and minor bone erosion—stratifying tumors from T1 (0 features) to T3 (≥4 features) [74]. BWH T2b–T3 tumors have the highest rates of nodal metastasis and recurrence and often define the cohort considered “advanced” at presentation or during follow-up [12,74]. A complementary role in staging is played by imaging techniques. Given that the overall risk of metastasis in cSCC is relatively low, routine radiologic screening is not recommended. However, computed tomography (CT), magnetic resonance imaging (MRI), ultrasound, and positron emission tomography/computed tomography (PET/CT) can be employed in high-risk tumors (AJCC T3–T4 or BWH T2b–T3) to evaluate bone involvement, deep soft tissue invasion, perineural spread, or the presence of nodal and distant metastases [75]. CT has emerged as the most frequently used modality for nodal and osseous assessment, whereas MRI is superior for the evaluation of perineural spread and deep structures; ultrasound with fine-needle aspiration is mainly applied for the assessment of superficial lymph nodes, while PET/CT, despite its higher sensitivity for distant metastatic disease, is limited by high costs and relatively modest impact on patient management. In Europe, in particular, ultrasound is often the first-line modality for the evaluation of regional lymph nodes in high-risk cases [75]. Finally, sentinel lymph node biopsy (SLNB) has been proposed as a tool for the early detection of occult micrometastases in selected patients with high-risk cSCC. In high-risk cSCC, early sentinel lymph node biopsy follows standard melanoma-derived protocols, with peritumoral radiocolloid injection and preoperative lymphoscintigraphy to map the draining basin, followed by radioguided (±blue dye) excision of the sentinel node(s) for serial sectioning and immunohistochemical assessment. Retrospective studies and meta-analyses have demonstrated that the procedure is technically feasible, with low false-negative rates and generally mild complications [76]. Reported positivity rates range between 11% and 24%, especially in tumors classified as BWH T2b or higher, suggesting a potential prognostic benefit in selected subgroups.

### 3.4. Therapeutic Strategies in Advanced cSCC

Management of aCSCC relies on multidisciplinary decision-making that weighs oncologic control against function, cosmesis, and patient comorbidities, within a guideline framework still constrained by limited randomized evidence [77,78]. North American (American Academy of Dermatology, AAD) and European (European Association of Dermato-Oncology, EADO; European Dermatology Forum, EDF; European Society for Radiotherapy and Oncology, ESTRO; European Union of Medical Specialists, UEMS; European Academy of Dermatology and Venereology, EADV; European Organisation for Research and Treatment of Cancer, EORTC) recommendation converge on a risk-adapted pathway. The therapeutic backbone shifts to systemic treatment and/or radiotherapy, with surgery reserved for selected, non-curative intents.

#### 3.4.1. Surgery in Advanced cSCC

In aCSCC, the surgery is reserved for selected, non-curative intents [77,78]. Operative interventions are considered salvage or adjunctive: (i) conversion/salvage resections after response to systemic therapy or RT when an R0 becomes achievable with acceptable morbidity; (ii) debulking/palliation to control pain, bleeding, infection, malodor, or to facilitate wound care; and (iii) metastasectomy or therapeutic nodal dissection in rigorously selected oligometastatic or regionally confined disease, ideally after multidisciplinary review [77,78,79].

#### 3.4.2. Radiotherapy and Management of Regional Nodal Disease

RT is the principal alternative for inoperable laCSCC or when surgery would incur unacceptable morbidity, and it is a key adjunct after surgery in selected scenarios [79,80]. It is particularly useful for tumors located on the lip, eyelid, nose, ear, or scalp, or in elderly patients with significant comorbidities. A meta-analysis of 14 observational studies including 1018 tumors showed a pooled recurrence rate of 6.4% after radiotherapy, higher than surgery but still within acceptable limits for selected patients [81]. Fractionation is tailored to fitness (standard 60–70 Gy vs. hypofractionation in frail patients) with awareness of late effects [82]. Postoperative RT is indicated when residual disease (R1/R2) cannot be cleared and is commonly advised for extranodal extension or extensive perineural invasion; after R0 resection, its role remains debated and is generally reserved for very high-risk biology (e.g., BWH T2b/T3), with emphasis on locoregional control rather than proven survival benefit [78]. Regional nodal disease should be surgically addressed when operability allows, with the extent of dissection individualized by basin and burden; parotid metastasis often warrants superficial parotidectomy with selective neck dissection, as RT alone underperforms for survival [77,83]. Elective dissection in clinically node-negative patients is not supported [78]. Sentinel lymph node biopsy reports low false-negative rates and double-digit positivity in BWH T2b/T3 tumors but lacks demonstrated survival benefit and remains investigational [77]. In unresectable nodal disease, definitive chemoradiation is an option; for systemic dissemination, systemic therapy becomes central. Among local palliative options, electrochemotherapy (ECT) offers high objective and complete response rates for unresectable cutaneous or in-transit lesions with acceptable toxicity, making it valuable for symptom control and field reduction in frail patients, though durability beyond short follow-up remains uncertain [84]. This technique can complement systemic therapy or defer extensive surgery/RT in selected cases. Notably, both radiotherapy and several cytotoxic regimens can dynamically up-regulate PD-L1 on tumor and immune cells through DNA-damage- and cGAS–STING/IFN-driven pathways, providing a biological rationale for combined chemo-/radio-immunotherapy with PD-1/PD-L1 inhibitors [85,86]. In this setting, RT may act as an in situ vaccine by inducing immunogenic cell death, enhancing antigen presentation, and increasing trafficking and activation of tumor-specific T cells, whereas checkpoint blockade prevents PD-1/PD-L1-mediated adaptive resistance and sustains systemic antitumor immunity [85,86,87,88]. Preclinical data and early clinical series, including patients with locally advanced cSCC treated with concurrent cemiplimab and RT, suggest that such combinations can improve local control and occasionally induce abscopal responses, although optimal dose, fractionation, and sequencing remain to be defined in prospective trials [85].

#### 3.4.3. Non-Immunotherapy Systemic Options

Cytotoxic chemotherapy—historically cisplatin ± 5-fluorouracil—produces short-lived responses and significant toxicity, particularly in the elderly aCSCC population, and is now largely a palliative fallback when immunotherapy is contraindicated or has failed [87]. EGFR-directed therapy (cetuximab; panitumumab) achieves disease control in a subset, with median survivals typically around a year in small series, and may serve as a bridge or radiosensitizer in non-immunotherapy candidates; dermatologic toxicities are frequent and require proactive management [88,89]. EGFR tyrosine kinase inhibitors have modest activity and limited durability [77].

#### 3.4.4. Translational and Emerging Therapies

Translational strategies aim to deepen or rescue responses to PD-1 blockade. Oncolytic immunotherapy (e.g., RP1) seeks to convert immunologically “cold” tumors to “hot,” with phase III (CERPASS) and phase II programs exploring combinations with PD-1 inhibitors; early signals in PD-1–refractory cohorts are encouraging but not yet definitive for practice change [77]. Additional targeted or immune-modulating agents (e.g., PLK1 inhibition with rigosertib in RDEB-associated aCSCC) illustrate the heterogeneity of advanced disease and the need for tailored trials in genetically or clinically defined subsets [78].

### 3.5. Immune Checkpoint Inhibitors in Advanced cSCC

Immune checkpoint inhibitors (ICIs) have revolutionized the therapeutic landscape of several solid tumors, and their role in advanced cSCC has rapidly gained attention. The strong immunogenicity of cSCC, its high tumor mutational burden, and frequent PD-L1 expression provide a strong biological rationale for PD-1 blockade. In this context, cemiplimab has emerged as the first and currently only approved PD-1 inhibitor in Europe for patients with unresectable locally advanced or metastatic cSCC. The following sections will explore the mechanistic basis of PD-1/PD-L1 inhibition, summarize the clinical trial and real-world evidence, and examine safety considerations with a focus on immune-related adverse events.

#### 3.5.1. Mechanism of Action

PD-1, a CD28-family inhibitory receptor on activated T cells, dampens TCR/CD28 signaling upon binding PD-L1/PD-L2 via SHP-1/2 recruitment to its ITSM motif, dephosphorylating CD3ζ, ZAP-70, and PI3K and suppressing Akt/mTOR/MAPK pathways [90,91]. PD-1 is also expressed on CD4^+^ T cells, regulatory T cells, B cells, and subsets of NK/NKT cells, where it contributes to broader immunoregulatory effects within the tumor microenvironment [90]. In tumors, chronic antigen exposure drives PD-1–high “exhausted” TILs with reduced proliferation, cytokines (IL-2, IFN-γ, TNF-α), and cytotoxicity [90,91]. cSCC, UV-driven and highly mutagenic, is immunogenic yet exploits adaptive resistance: IFN-γ–induced PD-L1 upregulation on tumor and myeloid cells attenuates antitumor T-cell activity, reinforced by Tregs, IL-10/TGF-β, M2 macrophages, and impaired type I IFN signaling [90,91,92,93]. The markedly higher cSCC incidence in immunosuppressed hosts (e.g., transplant recipients) underscores the role of immune surveillance. Anti-PD-1 antibodies (cemiplimab, pembrolizumab, nivolumab) block PD-1–ligand interactions, releasing the inhibitory brake and restoring antigen-specific T-cell function [94,95]. Mechanistically, PD-1 blockade reinvigorates exhausted CD8+/CD4+ T cells, enhancing proliferation, IFN-γ/TNF-α production, and perforin/granzyme-mediated cytolysis; additional effects include improved NK activity, revitalized tumor-draining node responses, and modulation of myeloid cells, with a possible, but generally outweighed, expansion of PD-1+ Tregs (Figure 4) [96].

The high tumor mutational burden and UV-signature neoantigens in cSCC likely underpin the robust clinical responses observed in advanced disease [90,93]. Baseline PD-L1 expression and pre-existing TILs correlate with higher response rates, reflecting the need for a pre-activated immune microenvironment for effective checkpoint blockade [92,93].

#### 3.5.2. Clinical Trials and Regulatory Approvals

cSCC is of great interest for treatment with ICIs. The high tumor mutational burden (TMB), defined as the number of nonsynonymous mutations per coding region, has been correlated to the response rate of PD1 antibody treatment [97]. In fact, high TMB increases the probability of generating neoantigens that can be recognized by tumor-infiltrating T cells, thereby enhancing the likelihood of a pre-existing antitumor immune response that can be unmasked by PD-1 blockade. In addition, immunohistochemical studies demonstrate relatively high expression of PD-L1 in cSCC, expressed not only on tumor cells but also on tumor-associated macrophages and other stromal elements. Increased PD-L1 expression has been shown to correlate with better tumor response to PD-1 antibody treatment, reinforcing the biological rationale for checkpoint blockade in this disease [98]. The pivotal clinical evidence comes from the EMPOWER-CSCC-1 trial, a phase II, non-randomized, multicohort study evaluating cemiplimab, a fully human IgG4 monoclonal antibody against PD-1 [99]. In the first cohort, 59 adult patients with metastatic disease and 78 with locally advanced tumors ineligible for curative surgery or radiotherapy received cemiplimab intravenously at 3 mg/kg every two weeks for up to 96 weeks. A separate cohort of 56 adult patients with metastatic cSCC was treated with a flat fixed dose of 350 mg intravenously every three weeks for up to one year. Of the total enrolled population, 33.7% had previously received systemic anticancer therapy. The primary endpoint was the overall response rate (ORR) as determined by independent central review using RECIST 1.1 criteria for radiographic assessment and modified WHO criteria for photographic evaluation [98]. Results presented at ESMO 2022, with a median follow-up of 15.3 months, reported an ORR of 50.8% in the 3 mg/kg group, which included 20.3% complete responses (CR) and 30.5% partial responses (PR) [74,100]. In the cohort with locally advanced disease, the ORR was 44.9%, with 12.8% CR and 32.1% PR. In the fixed-dose cohort, the ORR was 46.4%, comprising 19.6% CR and 26.8% PR. When all three groups were pooled (n = 193), the ORR was 47.2%, including 17.1% CR and 30.1% PR. The median duration of response reached 41.3 months, progression-free survival (PFS) was 22.1 months, and the median overall survival (OS) had not been reached at data cut-off. Treatment discontinuation due to adverse events occurred in 10.4% of patients. The most frequently observed toxicities were fatigue, diarrhea, nausea, and pruritus, consistent with the expected class profile. No treatment-related deaths were reported. Based on these data, cemiplimab was approved by the FDA in September 2018 and by the EMA in July 2019 for the treatment of patients with locally advanced or metastatic cSCC who are not candidates for curative surgery or radiotherapy [100]. A confirmatory cohort (group 6 of EMPOWER-CSCC-1) tested the regulatory-approved flat dosing regimen of cemiplimab at 350 mg every three weeks. At ESMO 2022, results from 167 patients (59.9% metastatic, 40.4% locally advanced) were presented, with a median treatment exposure of 35.7 weeks [101]. The median follow-up was 8.7 months. The overall response rate was 45.1%, including 5.5% complete responses. Median progression-free survival was 14.7 months, while the medians for both OS and duration of response had not yet been reached. Treatment discontinuation due to adverse events was reported in 13.9% of patients, with fatigue (26.1%), diarrhea (21.2%), and pruritus (21.2%) being the most common. No treatment-related deaths were observed in this cohort. Clinical trials of cemiplimab in advanced cutaneous squamous cell carcinoma are summarized in Table 3.

Pembrolizumab has also been evaluated in advanced cSCC, most notably in the KEYNOTE-629 trial [102]. This phase II study enrolled 105 patients with locally advanced or metastatic cSCC, the majority of whom had received prior systemic chemotherapy. The ORR was 34.3%, with 3.8% complete responses. Median PFS was 6.9 months, and median OS had not yet been reached, though 60.3% of patients were alive at 12 months. The discontinuation rate due to adverse events was 12.1%, and toxicity was comparable to that reported in cemiplimab studies. Subsequent analyses confirmed durable antitumor activity without unexpected safety concerns [103]. Pembrolizumab received FDA approval in June 2020 for locally advanced or metastatic cSCC but remains unapproved by EMA, leaving cemiplimab as the sole approved PD-1 inhibitor for advanced cSCC in Europe. Indirect comparative analyses have provided further evidence for the superior efficacy of cemiplimab. A systematic review by Keeping and colleagues evaluated 11 studies, including the cemiplimab EMPOWER-CSCC-1 trial (*n* = 193), seven trials of EGFR inhibitors, two pembrolizumab studies, and one platinum-based chemotherapy trial [104]. Cemiplimab demonstrated improved OS compared with EGFR inhibitors, with HRs ranging from 0.07 to 0.47, and PFS benefits with HRs between 0.30 and 0.67. Compared with pembrolizumab, cemiplimab conferred superior OS (HR range: 0.17–0.52) and PFS (HR range: 0.49–0.55 in KEYNOTE-629), though not in all datasets, as no significant PFS difference was observed in the Maubec et al. pembrolizumab trial [105]. Against platinum-based chemotherapy, cemiplimab showed a marked survival benefit (HR for OS: 0.19, 95% CI: 0.10–0.39), although the PFS advantage was not statistically significant (HR: 0.66, 95% CI: 0.38–1.16). A broader systematic review and comparative analysis by Petzold et al. included 22 studies evaluating chemotherapy, EGFR inhibitors, immune checkpoint inhibitors, and combination regimens [106]. Immunotherapy demonstrated the most favorable survival outcomes, with median PFS of 9.9 months (range 8.1–19.9) compared to 3 months for chemotherapy, 4.9 months for EGFR inhibitors, and 9.1 months for non-immunotherapy combination regimens. Median OS with checkpoint blockade was not reached (95% CI: 31.5 months–not reached), whereas it was 12.6 months for chemotherapy and 12.7 months for EGFR inhibition. At 26 months, survival with checkpoint inhibitors was 70.8% (95% CI: 61.5–81.5), compared with 17.1% (95% CI: 9.5–30.8) for chemotherapy and 37.9% (95% CI: 29.5–48.8) for combination regimens without immunotherapy.

Beyond advanced disease, the potential of cemiplimab in earlier treatment settings has been evaluated. The REGN-ONC-1901 neoadjuvant trial enrolled 79 patients with stage II–IV cSCC (median age 73 years) who received four cycles of cemiplimab 350 mg every three weeks prior to surgery [106]. Pathological complete response (pCR) was achieved in 56% of patients, with an additional 12.7% achieving major pathological response (1–10% viable tumor cells). By contrast, radiological ORR was 68.4%, including only 6.3% complete responses, underscoring the importance of pathological endpoints. PD-L1 expression ≥1% correlated with higher pCR rates, though this did not reach statistical significance. Four deaths occurred, three unrelated to study drug, and one discontinuation was reported. In the adjuvant setting, the REGN-1788 (C-POST) phase III trial is randomizing high-risk cSCC patients to cemiplimab (350 mg every three weeks for 12 weeks, followed by 700 mg every six weeks for 36 weeks) versus placebo after surgery and postoperative radiation [78]. The primary endpoint is disease-free survival. In parallel, the KEYNOTE-630 trial is assessing pembrolizumab (400 mg every six weeks for one year) versus placebo under similar conditions [78]. Alternative delivery strategies are also being tested. An intralesional phase I study (NCT03889912) administered cemiplimab directly into cutaneous lesions at doses ranging from 5 to 44 mg per lesion over 12 weeks prior to surgery [78,107]. Among 17 patients (median age 76 years, 76% with head and neck primaries), 76.5% achieved a pathological complete response. Pruritus (23.5%) and fatigue (17.6%) were the most common adverse events. One patient discontinued at the highest dose level, and no treatment-related deaths occurred. Combination approaches are under investigation as well. The I-TACKLE trial (NCT03666325) studied pembrolizumab monotherapy followed by the addition of cetuximab in patients with progression. Pembrolizumab alone produced an ORR of 44% (12 CR, seven PR) [108]. Among those who relapsed after initial response, two patients treated with pembrolizumab plus cetuximab achieved further partial responses. Among patients with no initial response to pembrolizumab, 38% responded after the addition of cetuximab. Parallel trials are ongoing to assess PD-1/PD-L1 inhibitors in combination with EGFR inhibitors, such as the ongoing study of avelumab with or without cetuximab (NCT03944941) [78].

#### 3.5.3. Real-World Evidence of Cemiplimab in Advanced cSCC

Beyond clinical trials, several real-world studies have provided valuable insights into the safety and efficacy of cemiplimab in patients with advanced cSCC, a population often older and more comorbid than those typically enrolled in pivotal trials. The CASE trial (NCT03836105), a prospective observational study, reported interim data on 196 patients with a median age of 76 years [109]. Most patients presented with locally advanced disease (63.3%), while 36.7% had metastatic cSCC; 42.9% had received prior radiotherapy, 75% prior surgery, and 45.4% prior systemic therapy. The overall response rate (ORR) was 37.4%, with 9.8% complete responses, and the disease control rate (DCR) reached 54.6%. Only 2.6% of patients discontinued treatment due to serious adverse events, and one death was attributed to treatment-related pneumonitis. In Europe, additional real-world cohorts have strengthened these findings. The CEMISKIN study, a large international multicentre non-interventional project conducted across Germany, Austria, and Switzerland, aims to include approximately 400 patients (200 prospective, 200 retrospective) to assess efficacy and tolerability of cemiplimab outside clinical trials [78,110]. Preliminary results highlight the feasibility of long-term data collection and confirm its integration into routine oncology practice. Similarly, the CAREPI study group in France analyzed 245 patients (mean age 77 years, 73% male), of whom 35% had locally advanced and 65% metastatic cSCC [111]. Nearly half had received prior systemic therapy, and 24% were immunocompromised. The best overall response rate was 50.4% (21% CR, 29% PR). With a median follow-up of 12.6 months, median progression-free survival (PFS) was 7.9 months, while median overall survival (OS) and duration of response were not reached. One-year OS was 73% in patients with ECOG < 2 versus 36% in those with ECOG ≥ 2, highlighting performance status as a critical prognostic factor. Severe treatment-related adverse events occurred in 9% of patients, including one fatal toxic epidermal necrolysis, while immunosuppression did not significantly impair efficacy. Data from the Netherlands have further validated these outcomes. In a retrospective multicentre cohort of 65 patients (median age 76, range 30–93), cemiplimab achieved an ORR of 52%, with 22% complete responses [112]. Median PFS was 10.9 months and median OS 26.1 months after a median follow-up of 21.5 months. Grade 3–4 adverse events occurred in 22% of patients, yet treatment was generally well tolerated, even among elderly patients with severe comorbidities. Complementary prospective data from the DRUG Access Protocol (151 patients treated with cemiplimab prior to national reimbursement) confirmed consistent efficacy: clinical benefit rate was 54.3%, ORR 35.1%, and median OS 24.2 months, with manageable safety [113]. Kidney transplant rejection emerged as the most notable treatment-related complication, affecting 9.5% of patients. Finally, a Hungarian single-centre retrospective analysis including 25 patients (median age 78 years) reported an ORR of 52% (3 CR, 10 PR) and a DCR of 76%, with similar efficacy observed in immunocompromised patients (ORR 60%, DCR 80%) [114]. Treatment discontinuation occurred in 24% due to adverse events, and 36% experienced grade ≥3 toxicities. Despite this, cemiplimab demonstrated clinically meaningful activity in a population with advanced age, multiple comorbidities, and frequent immunosuppression.

#### 3.5.4. Safety Profile and Immune-Related Adverse Events

Cemiplimab, as an anti–PD-1 antibody, shares with other ICIs the potential to induce a wide spectrum of immune-related adverse events (irAEs) [115]. PD-1 blockade enhances T-cell proliferation and effector function by preventing PD-1 ligation with PD-L1/PD-L2, but this same mechanism may precipitate autoreactive immune activity against healthy tissues. Breakdown of immune self-tolerance is mediated by clonal expansion of autoreactive T cells, diminished regulatory T-cell function, enhanced B-cell activation with autoantibody production, and proinflammatory cytokine release, which collectively underlie the onset of irAEs [116]. Organs dependent on peripheral tolerance, such as the skin, gastrointestinal tract, endocrine glands, and lungs, are most frequently affected. Clinical data from pivotal phase II trials highlight the incidence and severity of such events. In the open-label study of Migden et al., grade 3–4 treatment-emergent adverse events occurred in 44% of patients, with hypertension (8%) and pneumonia (5%) being most common [117]. Importantly, grade ≥ 3 immune-related events attributable to cemiplimab were reported in 10% of cases [117]. These included pneumonitis, hepatitis with marked transaminase elevation, proctitis, encephalitis, and autoimmune hepatitis, reflecting the multisystemic autoimmune profile of checkpoint blockade. Treatment discontinuation due to adverse events occurred in 8% of patients, and deaths were rarely but occasionally attributed to treatment, such as fatal pneumonia or aspiration pneumonia temporally related to therapy. Population-level real-world analyses have further refined the spectrum of adverse effects. In a retrospective global cohort of 714 cemiplimab-treated patients compared with almost 295,000 controls, significant excess risks were observed for several irAEs [118]. The strongest associations were seen for hypothyroidism (OR 8.06), rash or skin eruption (OR 7.02), elevated alkaline phosphatase (OR 6.54), hyponatremia (OR 5.26), diarrhea (OR 3.43), noninfectious colitis (OR 2.69), and pneumonia (OR 2.08). Notably, endocrine toxicities, especially thyroid dysfunction, were among the most frequent, consistent with other PD-1 inhibitors. Severe pulmonary complications were particularly clinically relevant: pneumonia and pleural effusion not only occurred more frequently but were also associated with a significant reduction in overall survival, with hazard ratios of 2.37 and 1.52, respectively [118].

These data emphasize the dual contribution of autoimmune and infectious etiologies in pulmonary irAEs, underscoring the importance of careful differential diagnosis. Cutaneous toxicities, including maculopapular rash and pruritus, are among the earliest and most common irAEs and were observed in up to 27% of patients in clinical trials [117]. Although generally mild (grade 1–2). Gastrointestinal adverse events, such as diarrhea and colitis, occurred in 7–8% of patients in real-world series and can progress to life-threatening immune-mediated colitis if unrecognized [118]. Hepatotoxicity, though less frequent, remains clinically significant, with elevations in AST/ALT reported in 20% of treated patients and rare cases of autoimmune hepatitis requiring discontinuation. Endocrine toxicities are dominated by thyroiditis leading to hypothyroidism, but rare cases of hypophysitis, adrenal insufficiency, and new-onset diabetes mellitus have also been described, consistent with PD-1–mediated disruption of glandular self-tolerance. Neurological and cardiovascular irAEs are rare but severe. Encephalitis, reported in phase II trials, as well as myocarditis and myositis, have been documented sporadically, carrying high mortality rates [117]. Renal involvement is uncommon but clinically relevant, with elevated creatinine and acute interstitial nephritis reported more frequently in cemiplimab-exposed patients, with renal failure occurring in approximately 20% of real-world cases, though often multifactorial [118]. Electrolyte disturbances such as hyponatremia and hypercalcemia are also more prevalent in cemiplimab recipients, suggesting systemic immune dysregulation.

Overall, the safety profile of cemiplimab in advanced cSCC mirrors that of other PD-1 inhibitors but is characterized by a predominance of cutaneous, endocrine, hepatic, and pulmonary irAEs. The majority are manageable with treatment interruption, corticosteroids, or immunosuppression, yet a subset of events, particularly pneumonitis, myocarditis, and severe hepatitis, carry substantial morbidity and mortality. Early recognition, patient selection, and vigilant monitoring, especially in elderly and comorbid populations typical of advanced cSCC, are therefore critical to optimizing outcomes.

## 4. Discussion

aCSCC remains a disease of high clinical and societal burden with notable health-system costs. In an analysis of healthcare resource utilisation within the Italian National Health Service, the mean annual cost per patient in 2015 was estimated at €5654 for non-melanoma skin cancers overall, rising to €10,281 for unresectable or advanced cSCC [67]. For decades, systemic options were largely palliative, offering transient control with substantial toxicity. The advent of PD-1 blockade has altered this landscape, introducing the possibility of durable benefit in a meaningful subset of patients. Yet, the key aspect for everyday practice is who benefits most, how durable that benefit is outside trials, and how to integrate systemic therapy with radiotherapy and selective surgery and these issues are far from settled.

In line with contemporary international guidelines, PD-1 blockade with cemiplimab, and, where available, pembrolizumab, is endorsed as the preferred first-line systemic option for most patients with unresectable or metastatic cSCC, given its favorable efficacy–toxicity profile and the potential for durable disease control [7,10,19,99]. Nonetheless, NCCN, ESMO and national societies (e.g., AIOM) explicitly acknowledge a role for alternative systemic strategies in selected scenarios, including EGFR inhibitors such as cetuximab and platinum-based chemotherapy, either as single agents or in combination regimens, particularly in patients with formal contraindications to immunotherapy (e.g., active autoimmune disease, uncontrolled interstitial lung disease), in those who progress on or shortly after PD-1 blockade, and in solid-organ transplant recipients where the risk of allograft rejection is deemed prohibitive [7,15]. Moreover, these agents may be integrated with radiotherapy and/or surgery in a multidisciplinary approach, for instance in the setting of bulky, borderline-resectable tumours or when rapid cytoreduction is required. Within this framework, cemiplimab should be regarded as the current backbone of systemic therapy for aCSCC rather than the sole available option, and its use needs to be contextualised within guideline-based, patient-tailored treatment algorithms.

Across pivotal trials and real-world cohorts, PD-1 blockade with cemiplimab has transformed the prognosis of advanced cSCC, with objective response rates (ORRs) in the range of ~45–60%, complete responses in 10–30% of patients, and median progression-free survival (PFS) that generally exceeds 8–12 months, alongside a tail of durable responders [110,111,112,113,114]. Nonetheless, outcomes remain highly heterogeneous, underscoring the need to understand host- and tumour-related prognostic factors to refine patient selection, counselling, and monitoring strategies. Some determinants have now been consistently validated, whereas others are emerging from broader immuno-oncology literature but remain insufficiently explored specifically in advanced cSCC treated with cemiplimab. Among clinicodemographic variables, performance status has emerged as one of the most powerful and reproducible prognostic determinants. In large real-world cohorts, patients with ECOG ≥ 1–2 systematically show lower ORR, shorter PFS, and reduced overall survival (OS), whereas those with good performance status achieve outcomes comparable to, or in some series approaching, those reported in EMPOWER-CSCC-1 [99]. Disease extent at baseline is another robust driver of prognosis. Pooled analyses of EMPOWER-CSCC-1 and multiple registries confirm that patients with locally advanced but non-metastatic disease derive the greatest relative benefit from cemiplimab, while nodal disease and, particularly, distant metastases are associated with lower response rates and shorter disease control, even though many metastatic cases still obtain meaningful benefit [99,110,111,112,113,114]. Immunosuppression status also carries substantial prognostic weight: real-world cohorts consistently report inferior PFS and OS in patients with solid-organ transplants, hematologic malignancies, chronic corticosteroid therapy, or other immunocompromising conditions, mirroring the historically more aggressive clinical course and poorer outcomes of cSCC in this setting [10,114]. By contrast, age and sex appear to exert limited influence on the probability of benefit. EMPOWER-CSCC-1 did not show meaningful differences in ORR or duration of response when stratified by age or gender, a finding that has been echoed in several elderly and ultra-octogenarian real-world series, supporting the use of cemiplimab irrespective of chronological age provided that functional status is acceptable [99,110,111,112,113,114]. Anatomical site is a more subtle variable. Cemiplimab series with adequate stratification suggest that head-and-neck primaries, which account for the majority of advanced cSCC, often display numerically higher response rates and more favorable survival than tumours on the trunk or extremities. In a large Italian multicenter cohort, head-and-neck localisation and normal baseline haemoglobin were significantly associated with better response, whereas genital primaries, prior chemotherapy, recent systemic antibiotics and chronic corticosteroid use correlated with poorer outcomes [10]. More recently, a multi-institutional UK study found that a head-and-neck primary site independently predicted improved PFS and OS, while concomitant immunosuppression portended worse PFS [119]. These observations are biologically plausible: chronic ultraviolet exposure on sun-exposed skin drives a very high tumour mutational burden and abundant neoantigens, and head-and-neck cSCC often exhibits increased PD-L1 expression and an inflamed tumour microenvironment, which may enhance susceptibility to PD-1 blockade [110,111,112,113,114]. At the same time, tumours on the trunk or extremities frequently arise in the context of chronic ulcers, lymphedema, diabetes or prior radiation, all of which can compromise local immune surveillance. Histopathology adds an additional layer of complexity. Surgical series and risk-stratification guidelines have long recognized that certain histologic variants, such as desmoplastic, spindle-cell, sarcomatoid, keratoacanthoma-like, and other poorly differentiated forms, are associated with higher rates of perineural invasion, local recurrence and disease-specific mortality compared with conventional cSCC [71,120,121]. These variants tend to display epithelial–mesenchymal transition, dense desmoplastic stroma and an immune-excluded phenotype, features that conceptually may reduce sensitivity to PD-1 blockade. However, most cemiplimab trials and registries have not systematically analysed outcomes by histologic subtype, and available data on variant histotypes under immunotherapy remain sparse and largely exploratory. Similarly, classical “high-risk” pathological features such as depth of invasion, lymphovascular invasion and perineural invasion, that are strong predictors of nodal spread and mortality in surgical cohorts, have not emerged as a consistent or strong prognostic signal in the limited cemiplimab datasets where they were captured, raising the possibility that checkpoint inhibition may partially mitigate their adverse impact. This highlights an important knowledge gap and supports systematic histopathologic stratification, including variant subtypes and quantitative measures of thickness and perineural involvement, in future immunotherapy studies. Beyond tumour burden and histology, host metabolic and inflammatory status is emerging as a potentially relevant, yet under-studied, prognostic domain in advanced cSCC. A growing body of pan-cancer evidence supports the so-called “obesity paradox”, whereby overweight or mildly obese patients (BMI ≥ 25 kg/m^2^) treated with ICIs often experience improved PFS and OS compared with normal-weight individuals, particularly in melanoma, NSCLC and head-and-neck cancers [122,123]. Proposed mechanisms include the immune-endocrine role of adipose tissue, leptin-driven T-cell activation coupled with PD-1 overexpression and functional exhaustion, and adiposity-associated changes in myeloid and lymphoid compartments that create a checkpoint-dependent state in which PD-1 blockade may yield a disproportionate functional gain [124,125]. However, BMI and other lifestyle-related variables (physical activity, smoking and alcohol intake, diet, UV exposure patterns) have been scarcely examined as prognostic modifiers in cemiplimab-treated cSCC, despite the high prevalence of elderly, comorbid and metabolically fragile patients in this population. In parallel, easily available serum biomarkers of systemic inflammation and nutritional status, albumin, neutrophil-to-lymphocyte ratio (NLR), and lactate dehydrogenase (LDH), have repeatedly been linked to ICI outcomes in other tumour types. Meta-analyses show that hypoalbuminemia is consistently associated with poorer survival in patients receiving ICIs, likely integrating systemic inflammation, cancer-associated cachexia and altered monoclonal antibody pharmacokinetics [126,127]. Elevated or rising NLR and dynamic increases in LDH during immunotherapy similarly correlate with treatment resistance and early progression in melanoma, NSCLC and head-and-neck carcinoma, whereas baseline values are less consistently predictive [128,129]. Despite their practicality and biological plausibility, these markers have only sporadically been incorporated into cSCC registries, and no validated threshold or composite score currently guides clinical practice in this setting.

Finally, the development of irAEs appears to be a promising “on-treatment” prognostic signal. Across multiple tumour types and ICI regimens, the occurrence of (mostly low-grade) irAEs, particularly cutaneous and endocrine, has been associated with higher response rates and prolonged survival, whereas severe toxicities requiring prolonged high-dose steroids may attenuate this benefit [129,130]. Evidence in cSCC specifically is still limited and partly conflicting, yet the biological rationale, e.g., shared T-cell clonality and cytokine profiles driving both antitumour and autoimmune responses, potential microbiome-mediated effects, and pharmacokinetic differences in non-cachectic patients, is compelling and justifies prospective collection and time-dependent analysis of irAEs as candidate surrogate markers of benefit with cemiplimab. Taken together, the available literature suggests a layered prognostic architecture in advanced cSCC treated with PD-1 blockade. “Core” determinants such as performance status, disease stage, immunosuppression, and (probably) anatomical site are now reasonably well characterized and should form the backbone of any risk stratification framework. In contrast, several potentially informative domains remain underexplored in this disease: histologic variants and quantitative pathologic features, lifestyle and metabolic factors (including BMI and related comorbidities), and inexpensive inflammatory or nutritional biomarkers (albumin, NLR, LDH and related indices), together with the timing, pattern and severity of irAEs.

Beyond traditional clinico-pathologic factors, emerging AI- and imaging-based approaches may further refine risk stratification in cSCC. Deep learning models applied to digital pathology and cellular morphometry have shown improved prediction of metastatic risk compared with conventional staging, suggesting a potential role in identifying patients who might benefit from intensified surveillance or systemic therapy [131,132]. Similarly, baseline [^18^F]FDG PET/CT radiomics signatures combined with AI are being explored to predict immunotherapy response and non-invasively approximate tumor grade in advanced cSCC. In parallel, hyperspectral imaging coupled with machine learning has demonstrated encouraging performance for non-invasive characterization and margin assessment of non-melanoma skin cancers, including SCC, and could in principle be integrated into pre-operative planning or longitudinal monitoring of high-risk lesions [133]. However, these tools remain at an early, exploratory stage, and robust prospective validation in dedicated aCSCC cohorts is still lacking.

In this context, future prospective, biomarker-rich studies and large, AI-applications, and harmonized real-world registries should be specifically designed to evaluate these factors in multivariable models, integrating static baseline characteristics with dynamic on-treatment markers within a truly multidisciplinary framework. This is particularly relevant in dermatology, where clinical practice is increasingly shifting towards stratified, team-based care and a greater focus on special populations (such as immunosuppressed, very elderly, highly comorbid or metabolically fragile patients), who may display distinct risk–benefit profiles under PD-1 blockade [134,135]. Robust prognostic tools capable of both identifying these patients upfront and anticipating their trajectory of response during therapy would ultimately allow clinicians to tailor treatment intensity, monitoring and supportive measures to the individual context, thereby refining risk–benefit assessment and personalizing management in advanced cSCC treated with cemiplimab.

## 5. Conclusions

Based on pivotal trials and accumulating real-world evidence, PD-1 blockade with cemiplimab now represents the standard first-line systemic option for unresectable locally advanced or metastatic cutaneous squamous cell carcinoma and should be considered early for all eligible patients without major contraindications to immunotherapy. Prognosis is not determined by stage alone, but by the interplay of tumor-related (tumor burden, head and neck localization, perineural invasion, metastatic pattern), host-related (age, ECOG performance status, immunosuppression, comorbidities) and inflammatory or nutritional markers (e.g., LDH, NLR, albumin, BMI). These domains should be systematically assessed at baseline to guide treatment goals, set realistic expectations and support shared decision-making, particularly in frail elderly and immunosuppressed patients. Our synthesis supports viewing cemiplimab as the backbone of care in aCSCC, to be integrated with surgery and radiotherapy within a multidisciplinary pathway. In clinical practice, we recommend early referral to dedicated skin cancer boards, adoption of structured follow-up schedules and routine collection of simple blood-based biomarkers and imaging at predefined time-points to detect primary resistance and to identify long-term responders who may be candidates for treatment de-escalation or discontinuation. Future research should prioritize the prospective validation of composite prognostic scores and biomarker-driven studies testing intensified or combined approaches in biologically high-risk subgroups, thereby consolidating cemiplimab at the center of personalized, risk-adapted management algorithms in aCSCC.

## Figures and Tables

**Figure 1 biomedicines-13-03010-f001:**
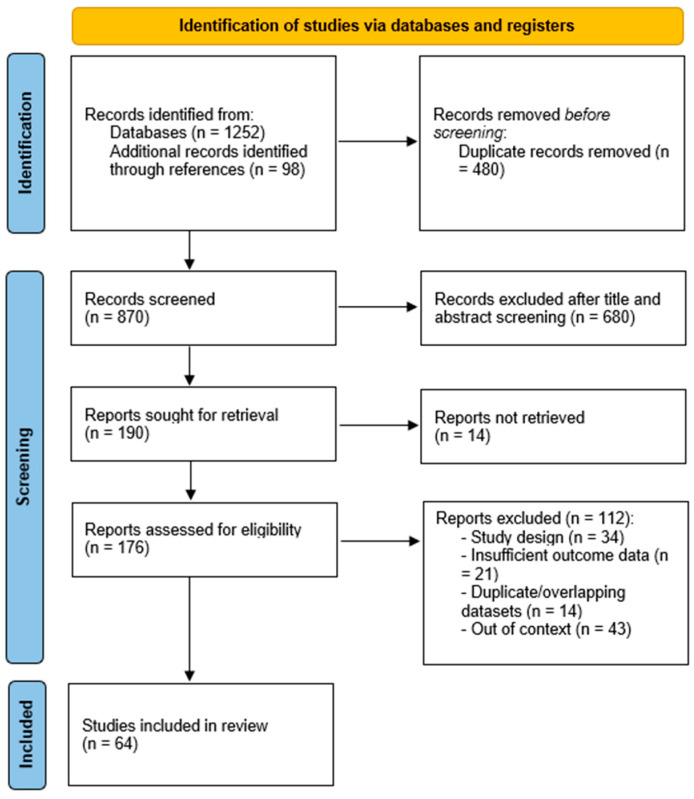
PRISMA flow diagram of study selection. Flow of records through identification, screening, eligibility assessment, and final inclusion of 64 studies in the narrative review on advanced cutaneous squamous cell carcinoma.

**Figure 2 biomedicines-13-03010-f002:**
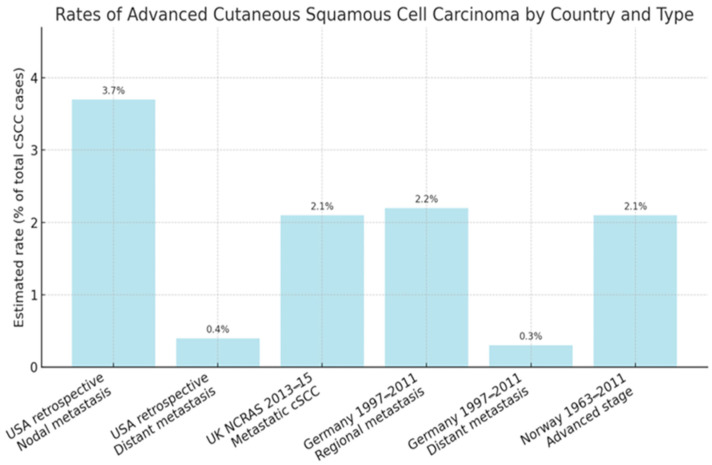
Estimated rates of advanced cSCC across different population-based studies. “Any advanced stage” includes both locally advanced and metastatic disease, where not otherwise specified. Data are derived from national cancer registry studies conducted in the United Kingdom (2013–2015), Germany (1997–2011), Norway (1963–2011), and epidemiological estimates from the United States (2012) [13,14,15,16,17].

**Figure 3 biomedicines-13-03010-f003:**
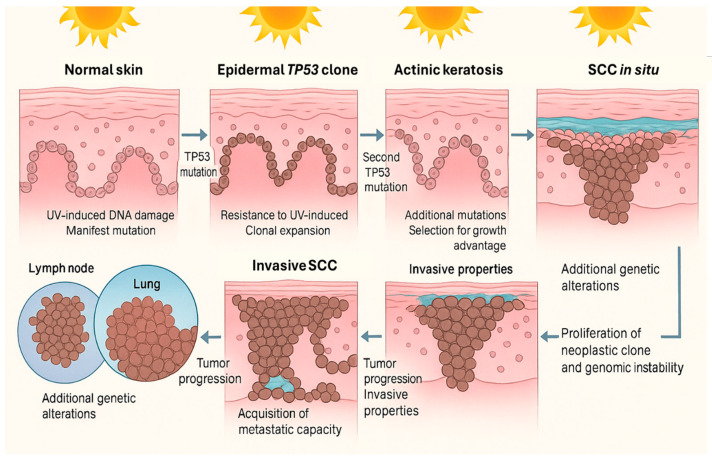
Multistep progression of cSCC. Chronic UV exposure leads to DNA damage and mutations in keratinocytes, including alterations in the *TP53* gene, allowing for clonal expansion and resistance to apoptosis. Accumulation of additional mutations promotes actinic keratosis and progression to squamous cell carcinoma in situ. Further genetic changes confer invasive and metastatic properties, enabling tumor spread to regional lymph nodes and distant organs. Image created using GNU Image Manipulation Program v.2.10.

**Figure 4 biomedicines-13-03010-f004:**
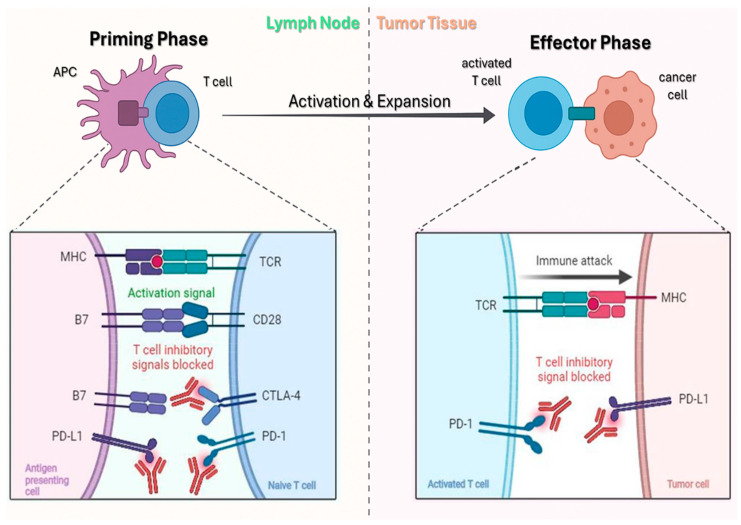
Mechanism of action of ICIs. Left: during the priming phase, APCs activate naïve T cells through MHC–TCR interaction, while inhibitory signals are blocked by anti–PD-1, anti–PD-L1, and anti–CTLA-4. Right: in the effector phase, activated T cells infiltrate tumor tissue, recognize tumor antigens, and trigger an immune attack, with inhibitory pathways blocked by anti–PD-1 and anti–PD-L1. Adapted from Khan et al. [96].

**Table 1 biomedicines-13-03010-t001:** Main pathogenetic and progression factors in cutaneous squamous cell carcinoma (cSCC).

Factor	Main Mechanisms	Impact on Advanced cSCC	References
Chronic UVR and therapeutic/ionizing radiation	UVB/UVA-induced DNA photoproducts with TP53 and other UV-signature mutations; PUVA and ionizing radiation add mutational burden	Drives field cancerization and accumulation of aggressive clones in chronically exposed skin	[25,28,30,31]
Chemical carcinogens	DNA adduct formation and impaired DNA repair (arsenic, PAHs, nitrosamines)	Cooperates with UVR in high-risk settings, favoring invasive and recurrent tumors	[32,33]
Systemic immunosuppression	Reduced immune surveillance; UV–drug synergy; carcinogenic immunosuppressants	65–250-fold higher incidence in SOTR and excess multifocal, rapidly progressive cSCC in CLL, NHL, HIV	[34,35,36,37]
β-HPV infection	E6/E7 interference with p53 and Rb in sun-exposed, predisposed or immunosuppressed hosts	Acts as co-carcinogen in early tumorigenesis, facilitating malignant transformation	[38,39]
Chronic inflammation, scars, non-healing wounds	Persistent ROS, cytokines and matrix remodeling in damaged tissue	Promotes Marjolin-type ulcers with aggressive local behavior	[40]
Cytokine/growth factor dysregulation	Aberrant TGF-β and IL-18/IL-37 signaling	Fosters invasion and immune escape, also in mucosal SCC	[41,42]
Oncogenic and iatrogenic drugs	Paradoxical MAPK activation under BRAF inhibitors; long-term voriconazole and vismodegib exposure	Eruptive keratoacanthomas/cSCC and acceleration of lesions in UV-damaged fields	[43,44]
Lifestyle factors	Smoking, alcohol, and occupation-related exposure patterns	Modestly increase risk and may amplify other carcinogenic drivers	[45]
Inherited genodermatoses	Defective *NER*, pigmentary and viral susceptibility disorders, chronic-wounding syndromes	Very high lifetime risk of multiple, early-onset and often advanced cSCC	[46,47,48]
Common germline variants	GWAS loci in pigmentation, immunity and keratinocyte regulation	Modulate baseline susceptibility and threshold for progression	[49,50]
Somatic driver alterations and field cancerization	*TP53*, *CDKN2A*, *NOTCH1/2, RAS, KNSTRN, MYC*, epigenetic changes	Orchestrate transition from UV-damaged fields to invasive and metastatic disease	[51,52,53]
Tumor microenvironment and immune escape	VEGF-driven angiogenesis, MHC-I loss, immunosuppressive cells/cytokines, PD-L1 expression	Consolidates immune evasion, invasion, metastasis and shapes response to ICI	[54,55]

Abbreviations: cSCC, cutaneous squamous cell carcinoma; SCC, squamous cell carcinoma; UVR, ultraviolet radiation; UVB, ultraviolet B; UVA, ultraviolet A; PUVA, psoralen plus UVA; PAHs, polycyclic aromatic hydrocarbons; SOTR, solid-organ transplant recipients; CLL, chronic lymphocytic leukemia; NHL, non-Hodgkin lymphoma; HIV, human immunodeficiency virus; HPV, human papillomavirus; ROS, reactive oxygen species; TGF-β, transforming growth factor-β; IL, interleukin; MAPK, mitogen-activated protein kinase; NER, nucleotide excision repair; GWAS, genome-wide association studies; CN, copy number; MHC-I, major histocompatibility complex class I; PD-L1, programmed cell death ligand 1; ICI, immune checkpoint inhibitor.

**Table 2 biomedicines-13-03010-t002:** Major risk factors for metastasis in invasive cSCC, including clinical, anatomical, and histopathologic parameters associated with aggressive behavior.

Risk Factor	High-Risk Criteria	Notes	References
Tumor thickness	>2 mm (especially >6 mm)	Greater thickness strongly correlates with metastatic potential	[1,6,12,67,68]
Diameter	>2 cm	Larger diameter increases risk of spread	[1,6,12,67,68]
Location	Ear, lips, mucosae (tongue, vulva, penis); perineural growth may increase risk	Certain anatomical sites have higher metastatic rates	[64,65,66,67]
Arising within a scar	Burn or radiation scar	Chronic scars are associated with aggressive behaviour	[65]
Histopathologic features	Poorly differentiated/undifferentiated; acantholytic type; arising in Bowen disease	Histologic variants with higher propensity for metastasis; acantholytic type risk has been recently questioned	[12,67,68]
Immunosuppression	Any cause of systemic immunosuppression	Includes transplant recipients, HIV infection, long-term immunosuppressive therapy	[64,68]

Abbreviations: cSCC, cutaneous squamous cell carcinoma; mm, millimeters; cm, centimeters; HIV, human immunodeficiency virus.

**Table 3 biomedicines-13-03010-t003:** Clinical trials exploring cemiplimab in advanced cSCC.

Study/Year	Trial Design, Median Follow-Up	Patients (N)	Disease Type	Treatment Schema	Response Outcomes	Survival Outcomes
Migden, 2018 [99]	Phase I, open-label, multicentre; 11 mo	26	10 locally advanced, 8 regional metastasis, 8 distant metastasis	Cemiplimab 3 mg/kg IV every 2 weeks	ORR 50% (13 PR)	Median duration of response not reached; OS/PFS not reported
Migden, 2018 [99] (Group 1)	Phase II, non-randomised, pivotal study; 7.9 mo	59	14 regional metastasis, 45 distant metastasis	Cemiplimab 3 mg/kg IV every 2 weeks	ORR 47.5% (4 CR, 24 PR)	1-yr OS: est. 81%; 1-yr PFS: est. 53%; median DOR not reached
Rischin, 2020 [101]	Phase II, follow-up; 16.5 mo	–	–	–	ORR 49% (10 CR, 19 PR)	Median PFS not reached; median DOR not reached
Rischin, 2021 [101]	Phase II, updated follow-up; 18.5 mo	–	–	–	ORR 51% (12 CR, 18 PR)	Median DOR not reached
Rischin, 2020 [101] (Group 3)	Phase II, non-randomised, fixed-dose cohort; 8.1 mo	56	12 regional metastasis, 43 distant metastasis	Cemiplimab 350 mg IV every 3 weeks	ORR 41% (3 CR, 20 PR)	Median PFS not reached; median DOR not reached
Rischin, 2021 [100] (Group 3, update)	Phase II, updated follow-up; 17.3 mo	–	–	Cemiplimab 350 mg IV every 3 weeks	ORR 42.9% (9 CR, 15 PR)	Median DOR not reached
Migden, 2020 [101] (Group 2)	Phase II, non-randomised, pivotal study; 9.3 mo	78	78 locally advanced	Cemiplimab 3 mg/kg IV every 2 weeks	ORR 44% (13% CR, 31% PR)	1-yr DFS: 87.8%; median DOR not reached
Rischin, 2021 [100] (Group 2, update)	Phase II, updated follow-up; 15.5 mo	–	Locally advanced	Cemiplimab 3 mg/kg IV every 2 weeks	ORR 44.9% (10 CR, 25 PR)	Median DOR not reached

Abbreviations: CR, complete response; PR, partial response; ORR, overall response rate; DOR, duration of response; PFS, progression-free survival; OS, overall survival; DFS, disease-free survival; mo, months; IV, intravenous.

## Data Availability

No new data were created or analyzed in this study. Data sharing is not applicable to this article.

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
