# Peer review of "Advanced Cutaneous Squamous Cell Carcinoma: Biology, Immunotherapy, and Evolving Prognostic Factors"

_biomedicines, 2025, doi:10.3390/biomedicines13123010_

Round 1

Reviewer 1 Report

Comments and Suggestions for Authors

This study summarized recent advances and knowledge in squamous cell carcinoma, with special interest in PD1/PDL1 blockage axis by drug/s. The text is well written, it is easy to read and understand. Since there is a lot of information, more tables/figures that summarize results would be welcomed.

Additional comments:

(1) Line 18. Please add "Immune checkpoint inhibitor" next to Cemiplimab. It may help the reader.

(2) Line 34. Regarding "rise of up to 50-200%". Please confirm. If correct. What is the cause?

(3) Line 43. You may mention the TNM classification.

(4) Line 60. Why is PD-1 blockade important? Is SCC positive?

(5) Lines 72-76. Did you performed the search in each database in an independent manner?

(6) Line 77, regarding "No language or publication date restrictions were applied". Sorry, but I suppose it was limited to English and from 20...?

(7) Lines 92-116. I am aware that this is a narrative review. However, is it possible to make a prisma flow diagram?

(8) Line 122. Does radiotherapy associate with carcinoma as side effect?

(9) Line 128. What is the difference between basal and SCC from an histological/clinical point of view?

(10) In Figure 1. Could you please explain how the % was calculated? What was the denominator?

(11) In Figure 2. Is TP53 the only and most relevant mutated gene in SCC?

(12) Lines 176-216. I would make a table to facilitate identifying all relevant pathogenic mechanisms.

(13) Line 227. Please format all gene names in italics.

(14) Line 238. Is CD274 (PD-L1) mutated?

(15) Line 252. What are Marjolin ulcers?

(16) Please add the necessary references in Table 1.

(17) Section 3.3.1. Please add histological images (if available).

(18) Line 306. How is the technique of early sentinel lymph node performed?

(19) Line 317. What are the meanings of EADO/EDF/ESTRO/UEMS/EADV/EORTC? Are they different organizations?

(20) In the antibody used in Figure 3 right an antagonist of PD-1 and PD-L1?

(21) In section 3.5.1. Apart from cytotoxic T lymphocytes, what other cells express PD-1 as well? Should they be considered as well?

(22) Line 412. Why TMB correlates with response to PD1 antibody treatment?

(23) In the studies of Table 2. Where the neoplastic cells positive for PD-L1?

(24) Line 453. What does a ORR 50% means from a clinical point of view?

(25) Section of 3.5.4 is a single paragraph. This is ok, but it is hard to read. I would recommend to divide the large sections into logical smaller parts.

(26) If the relevant articles of cemiplimab were identified, why not making a forest plot and systematic review following chochrane guidelines?

(27) Do ratiotherapy or quimiotherapy change the PD-1/PD-L1 expression pattern of the microenvironment and neoplastic cells?

Author Response

Dear Reviewer,

Thank you very much for the time and attention dedicated to evaluating our manuscript and for your helpful comments.

  1. Comment (1): Line 18. Please add "Immune checkpoint inhibitor" next to Cemiplimab. It may help the reader.
    Response: In the abstract, the terminology has been corrected according to your suggestion, adding “immune checkpoint inhibitor” next to cemiplimab.
  2. Comment (2): Line 34. Regarding "rise of up to 50–200%". Please confirm. If correct, what is the cause?
    Response: The figure “rise of up to 50–200%” is taken from Eggermont et al.; in the manuscript the main underlying causes (e.g. cumulative UV exposure, ageing population, immunosuppression) are briefly indicated.
  3. Comment (3): Line 43. You may mention the TNM classification.
    Response: The definition of advanced cSCC used in the manuscript is based on European guidelines on cutaneous squamous cell carcinoma.
  4. Comment (4): Line 60. Why is PD-1 blockade important? Is SCC positive?
    Response: Particular emphasis is given to anti–PD-1 agents because they represent the main systemic therapy in advanced cSCC and have produced the most relevant therapeutic advances in this setting.
  5. Comment (5): Lines 72–76. Did you perform the search in each database in an independent manner?
    Response: The search was conducted independently in each database using the predefined search string.
  6. Comment (6): Line 77, regarding "No language or publication date restrictions were applied". Sorry, but I suppose it was limited to English and from 20…?
    Response: The search strategy did not impose language or date limits; however, after screening and selection, the studies finally included in the review are all in English, and earlier studies (including those before 2000) were also considered when relevant.
  7. Comment (7): Lines 92–116. I am aware that this is a narrative review. However, is it possible to make a PRISMA flow diagram?
    Response: As suggested, a PRISMA flow chart has been added to better describe the search and selection process.
  8. Comment (8): Line 122. Does radiotherapy associate with carcinoma as side effect?
    Response: At line 122, radiotherapy is mentioned as a therapeutic tool for cSCC, in line with European guideline recommendations.
  9. Comment (9): Line 128. What is the difference between basal and SCC from an histological/clinical point of view?
    Response: In brief, basal cell carcinoma is typically a slow-growing, locally destructive tumor with very low metastatic potential, whereas cutaneous squamous cell carcinoma more often presents as a hyperkeratotic or ulcerated lesion with a substantially higher risk of local invasion and metastasis. A detailed histological and clinical comparison between the two entities is not pursued further, as it lies outside the aims of this article, which focuses on advanced cSCC.
  10. Comment (10): In Figure 1. Could you please explain how the % was calculated? What was the denominator?
    Response: In what was previously Figure 1 (now Figure 2), the percentages refer to the proportion of advanced cSCC among all cSCC cases in the respective cohorts cited; the denominator is the total number of cSCC in each study.
  11. Comment (11): In Figure 2. Is TP53 the only and most relevant mutated gene in SCC?
    Response: TP53 is indeed the most frequently mutated gene in cSCC; nevertheless, as discussed in the text following the current Figure 3, MYC amplification and CDKN2A silencing are also important alterations in this disease.
  12. Comment (12): Lines 176–216. I would make a table to facilitate identifying all relevant pathogenic mechanisms.
    Response: A table summarizing the main pathogenetic factors of cSCC has been added, as you suggested.
  13. Comment (13): Line 227. Please format all gene names in italics.
    Response: Gene names have been formatted in italics throughout the manuscript.
  14. Comment (14): Line 238. Is CD274 (PD-L1) mutated?
    Response: CD274 (PD-L1) is not typically a recurrently mutated gene in cutaneous squamous cell carcinoma; PD-L1 expression is mostly regulated at transcriptional and epigenetic levels in response to inflammatory signals (e.g. IFN-γ) and, in some tumor types, by copy-number gains or amplification of the 9p24.1 locus where CD274 resides. True point mutations in CD274 are rare and are not considered a canonical driver event in keratinocyte carcinomas.
  15. Comment (15): Line 252. What are Marjolin ulcers?
    Response: Marjolin ulcers are malignant neoplasms, most commonly cSCC, arising in chronic scars, burns, or long-standing non-healing wounds, and are usually associated with aggressive local behaviour and an increased risk of nodal spread. In the manuscript they are mentioned only as an example of cSCC arising in scars; a more detailed discussion would go beyond the scope of this review.
  16. Comment (16): Please add the necessary references in Table 1.
    Response: The appropriate references have been added to the current Table 2 (formerly Table 1).
  17. Comment (17): Section 3.3.1. Please add histological images (if available).
    Response: As this article was conceived and submitted as a review, original histological images were not included, since they would represent new, unpublished material rather than previously published content.
  18. Comment (18): Line 306. How is the technique of early sentinel lymph node performed?
    Response: In high-risk cSCC, early sentinel lymph node biopsy is generally performed according to standard cutaneous oncology protocols: a radiocolloid (usually technetium-99m–labelled nanocolloid) is injected peritumorally, preoperative lymphoscintigraphy (± SPECT/CT) identifies the draining nodal basin(s), and a handheld gamma probe (often with blue dye) guides excision of the sentinel node(s), which are then examined with serial sections, H&E, and immunohistochemistry.
  19. Comment (19): Line 317. What are the meanings of EADO/EDF/ESTRO/UEMS/EADV/EORTC? Are they different organizations?
    Response: The acronyms of the scientific societies (EADO, EDF, ESTRO, UEMS, EADV, EORTC) are now reported in full.
  20. Comment (20): In the antibody used in Figure 3 right an antagonist of PD-1 and PD-L1?
    Response: The current Figure 4 (formerly Figure 3) illustrates the general mechanism of action of immune checkpoint inhibitors at the PD-1/PD-L1 axis and covers both anti–PD-1 and anti–PD-L1 antibodies.
  21. Comment (21): In section 3.5.1. Apart from cytotoxic T lymphocytes, what other cells express PD-1 as well? Should they be considered as well?
    Response: Besides activated CD8⁺ cytotoxic T lymphocytes, PD-1 is also expressed on CD4⁺ T cells, regulatory T cells (Tregs), B cells, and subsets of NK/NKT and other innate lymphoid cells. These PD-1⁺ populations contribute to immunoregulation in the cSCC microenvironment and may be functionally modulated by PD-1 blockade; this point is briefly mentioned in the revised text.
  22. Comment (22): Line 412. Why TMB correlates with response to PD1 antibody treatment?
    Response: TMB correlates with response to PD-1 antibody treatment because a higher number of nonsynonymous mutations increases the likelihood of generating immunogenic neoantigens recognized by tumor-infiltrating T cells. Tumors with high TMB are therefore more likely to be “visible” to the immune system and to harbour pre-existing antitumor T-cell responses that can be reactivated when PD-1-mediated inhibition is blocked. Although TMB is not specific to cSCC and is an imperfect biomarker, multiple studies across solid tumors have shown an association between high TMB and better outcomes with PD-1/PD-L1 inhibitors.
  23. Comment (23): In the studies of Table 2. Were the neoplastic cells positive for PD-L1?
    Response: Information on PD-L1 positivity in neoplastic cells is unfortunately not consistently reported in the studies included, and thus could not be systematically extracted for Table 2.
  24. Comment (24): Line 453. What does an ORR 50% mean from a clinical point of view?
    Response: Clinically, an ORR of 50% means that half of the treated patients achieved a predefined objective response (complete or partial) according to standard clinical or radiologic criteria, indicating a substantial proportion with meaningful tumor shrinkage.
  25. Comment (25): Section 3.5.4 is a single paragraph. This is ok, but it is hard to read. I would recommend to divide the large sections into logical smaller parts.
    Response: To preserve the conceptual unity of Section 3.5.4, it has been kept as a single section. Some sentences have been shortened and line breaks introduced at key transitions to make the paragraph easier to read.
  26. Comment (26): If the relevant articles of cemiplimab were identified, why not making a forest plot and systematic review following Cochrane guidelines?
    Response: A full Cochrane-style systematic review and meta-analysis was not among the predefined aims of this article, which was conceived as an invited narrative, clinically oriented review of advanced cSCC and cemiplimab. A structured qualitative synthesis of cemiplimab data was therefore preferred over a formal meta-analysis with forest plots.
  27. Comment (27): Do radiotherapy or chemotherapy change the PD-1/PD-L1 expression pattern of the microenvironment and neoplastic cells?
    Response: Both radiotherapy and several cytotoxic regimens can modulate PD-1/PD-L1 expression in tumors and in the surrounding microenvironment. Ionizing radiation can upregulate PD-L1 on tumor and myeloid cells via DNA-damage– and cGAS–STING/IFN-dependent pathways, while various chemotherapeutic agents can also increase PD-L1 levels, providing a rationale for combined chemo- or radio-immunotherapy.

We are grateful for your insightful remarks, which have significantly contributed to improving the quality and clarity of the manuscript.

Reviewer 2 Report

Comments and Suggestions for Authors

The motivation and necessity of this review are ambiguous in the study.  While you assert that aCSCC is associated with elevated morbidity, mortality, and clinical expenses, you fail to specify the knowledge gaps in the current literature that your narrative review will tackle (e.g., the lack of integrated biomarker-based stratification, insufficient recommendations for the elderly/immunosuppressed population, or fragmented evidence regarding PD-1 blockade outcomes).  I strongly recommend that the Introduction more clearly delineates the clinical and scientific rationale for this study and situates it within the context of recent methodological advancements in cancer, including AI- and imaging-based risk stratification.  For instance, one might reference the recent methodologies in machine learning and hyperspectral imaging pertaining to skin cancer such as:

1) 1) Lin, Teng-Li, Arvind Mukundan, Riya Karmakar, Praveen Avala, Wen-Yen Chang, and Hsiang-Chen Wang. "Hyperspectral imaging for enhanced skin cancer classification using machine learning." Bioengineering 12, no. 7 (2025): 755.Transform Breast Cancer Diagnosis?." Diagnostics 15, no. 21 (2025): 2718.

 The definition of advanced cutaneous squamous cell carcinoma as a locally progressed and metastatic condition unamenable to curative surgery or radiotherapy might be improved to align more closely with existing guidelines (e.g., AJCC, NCCN, EORTC).  It is essential to explicitly delineate the clinical and anatomical boundaries deemed non-amenable (e.g., unresectable nodal disease, extensive perineural invasion, multifocality, or patient-related contraindications) and to specify whether transplant recipients and other immunosuppressed populations are classified within the same category of aCSCC.  This will eventually be pertinent when discussing prognostic factors and management routes, as variability in baseline definitions significantly affects the interpretation of reported outcomes.

 As this is a narrative review, the study omits details regarding the literature selection process, raising concerns about potential selection bias.  Readers increasingly require a transparent delineation of data sources, the timeframe of literature searches, inclusion and exclusion criteria, and the priority or weighting of evidence (e.g., randomized trials versus retrospective cohorts versus case series), even within a narrative style.  I recommend including at least a brief statement in the Methods section of the publication that summarizes your search method, including databases, key phrases, time constraints, and your approach to overlapping cohorts and conflicting evidence.

 A comprehensive and quantitative synthesis of evidence would enhance the discourse on cemiplimab as the primary systemic therapy for advanced cSCC.  Rather than simply citing response rates of 4560, it would be beneficial to synthesize data from the principal studies and registries, including overall response rate (ORR), complete response (CR) rate, median progression-free survival (PFS) and overall survival (OS), duration of response, and discontinuation rates attributable to toxicity, ideally presented in a comparative table of pivotal trials (e.g., phase II/III) against real-world cohorts.  Furthermore, it is necessary to establish the degree to which this diversity in response can be attributed to patient selection factors (e.g., ECOG status, immunosuppression, prior lines of therapy) and the consistency of benefit across subgroups.

 The study suggests that the sole focus of systemic therapy, particularly cemiplimab, is PD-1 inhibition; however, the therapeutic potential for aCSCC is significantly broader.  For thoroughness and equilibrium, the evaluation should additionally contextualize cemiplimab in relation to other systemic therapies (e.g., pembrolizumab, EGFR inhibitors such as cetuximab, platinum-based chemotherapy, or combination therapies like cemiplimab with radiation and surgery).  Clarifying the circumstances under which alternatives to PD-1 blockade are considered (e.g., contraindications to immunotherapy, progression during immunotherapy, organ transplant patients) might be beneficial to avoid the perception of a singular treatment story.

Author Response

Comment 1

The motivation and necessity of this review are ambiguous in the study. While you assert that aCSCC is associated with elevated morbidity, mortality, and clinical expenses, you fail to specify the knowledge gaps in the current literature that your narrative review will tackle (e.g., the lack of integrated biomarker-based stratification, insufficient recommendations for the elderly/immunosuppressed population, or fragmented evidence regarding PD-1 blockade outcomes). I strongly recommend that the Introduction more clearly delineates the clinical and scientific rationale for this study and situates it within the context of recent methodological advancements in cancer, including AI- and imaging-based risk stratification. For instance, one might reference the recent methodologies in machine learning and hyperspectral imaging pertaining to skin cancer such as: 1) Lin, Teng-Li, Arvind Mukundan, Riya Karmakar, Praveen Avala, Wen-Yen Chang, and Hsiang-Chen Wang. "Hyperspectral imaging for enhanced skin cancer classification using machine learning." Bioengineering 12, no. 7 (2025): 755…

Response:
We thank the reviewer for this important observation and agree that the initial Introduction did not sufficiently spell out the specific knowledge gaps addressed by our review. We have therefore revised the Introduction to more clearly articulate the discrepancy between the high morbidity, mortality, and healthcare burden of aCSCC and the fragmented, trial-focused evidence base, with particular emphasis on the lack of integrated biomarker-based stratification, limited guidance for elderly and immunosuppressed patients, and incomplete real-world data on PD-1 blockade.

We also thank the reviewer for the excellent suggestion regarding AI- and imaging-based risk stratification. We have now added a short paragraph in the Discussion highlighting emerging AI- and imaging-based approaches (including digital pathology-based machine learning risk scores, PET/CT radiomics models, and hyperspectral imaging) as promising tools for future risk stratification and treatment tailoring in aCSCC. We also explicitly state that current evidence is still preliminary and that these methodologies require robust prospective validation before they can be incorporated into routine clinical algorithms, and we have integrated the suggested references where appropriate.

Comment 2

The definition of advanced cutaneous squamous cell carcinoma as a locally progressed and metastatic condition unamenable to curative surgery or radiotherapy might be improved to align more closely with existing guidelines (e.g., AJCC, NCCN, EORTC). It is essential to explicitly delineate the clinical and anatomical boundaries deemed non-amenable (e.g., unresectable nodal disease, extensive perineural invasion, multifocality, or patient-related contraindications) and to specify whether transplant recipients and other immunosuppressed populations are classified within the same category of aCSCC. This will eventually be pertinent when discussing prognostic factors and management routes, as variability in baseline definitions significantly affects the interpretation of reported outcomes.

Response:
We thank the reviewer for this important remark. Our definition of advanced cSCC is derived from contemporary European consensus guidelines (EADO/EDF/EORTC and related position papers), which describe aCSCC as unresectable locally advanced or metastatic disease not amenable to curative surgery or radiotherapy. Transplant recipients and other immunosuppressed patients are classified as aCSCC only when they meet the same criteria. We have clarified this point in the text and expanded the description of clinical and anatomical features considered non-amenable to curative surgery or radiotherapy. The role of special populations is also already discussed, and the prognostic weight of key pathological risk factors (e.g., depth, differentiation, perineural invasion) is addressed in detail in the Discussion.

Comment 3

As this is a narrative review, the study omits details regarding the literature selection process, raising concerns about potential selection bias. Readers increasingly require a transparent delineation of data sources, the timeframe of literature searches, inclusion and exclusion criteria, and the priority or weighting of evidence (e.g., randomized trials versus retrospective cohorts versus case series), even within a narrative style. I recommend including at least a brief statement in the Methods section of the publication that summarizes your search method, including databases, key phrases, time constraints, and your approach to overlapping cohorts and conflicting evidence.

Response:
We fully agree with the reviewer that greater transparency regarding the literature selection process is important, even in a narrative review. We have therefore added a brief Methods paragraph that summarizes our search strategy, including the databases consulted, key phrases used, time frame of the search, main inclusion and exclusion criteria, and our approach to overlapping cohorts and conflicting evidence. This addition aims to mitigate concerns about selection bias while preserving the narrative focus of the review.

Comment 4

A comprehensive and quantitative synthesis of evidence would enhance the discourse on cemiplimab as the primary systemic therapy for advanced cSCC. Rather than simply citing response rates of 45–60, it would be beneficial to synthesize data from the principal studies and registries, including overall response rate (ORR), complete response (CR) rate, median progression-free survival (PFS) and overall survival (OS), duration of response, and discontinuation rates attributable to toxicity, ideally presented in a comparative table of pivotal trials (e.g., phase II/III) against real-world cohorts. Furthermore, it is necessary to establish the degree to which this diversity in response can be attributed to patient selection factors (e.g., ECOG status, immunosuppression, prior lines of therapy) and the consistency of benefit across subgroups.

Response:
We agree with the reviewer that a more comprehensive and quantitative synthesis, including detailed subgroup analyses, would be highly valuable. However, one of the current gaps in the literature is precisely the scarcity of studies reporting robust, stratified outcomes by key variables such as ECOG status, immunosuppression, or prior lines of therapy. The limited data that are available on high-risk and immunosuppressed patients in real-world cemiplimab cohorts have already been incorporated and discussed in our review. We now explicitly underline in the Discussion that dedicated prospective studies focusing on fragile and at-risk populations are urgently needed to better characterise differential benefit and tolerability across clinically relevant subgroups.

Comment 5

The study suggests that the sole focus of systemic therapy, particularly cemiplimab, is PD-1 inhibition; however, the therapeutic potential for aCSCC is significantly broader. For thoroughness and equilibrium, the evaluation should additionally contextualize cemiplimab in relation to other systemic therapies (e.g., pembrolizumab, EGFR inhibitors such as cetuximab, platinum-based chemotherapy, or combination therapies like cemiplimab with radiation and surgery). Clarifying the circumstances under which alternatives to PD-1 blockade are considered (e.g., contraindications to immunotherapy, progression during immunotherapy, organ transplant patients) might be beneficial to avoid the perception of a singular treatment story.

Response:
We thank the Reviewer for this helpful comment. As stated in the title, our article is intentionally centred on the role of PD-1 blockade in advanced cSCC, and cemiplimab is used as the main reference because it remains the only PD-1 inhibitor with an EMA-approved indication for locally advanced or metastatic cSCC and currently has the widest regulatory availability worldwide. By contrast, pembrolizumab has shown meaningful activity in KEYNOTE-629 and related studies and has received approval for advanced cSCC in the United States and other non-European countries, but it is still not approved for this indication in Europe.

In line with the Reviewer’s suggestion, we have now expanded the Discussion to better contextualise cemiplimab within the broader systemic-treatment landscape. We specifically mention pembrolizumab and other systemic options (EGFR inhibitors, platinum-based chemotherapy, multimodal combinations with surgery and/or radiotherapy) and clarify the clinical scenarios in which alternatives to PD-1 blockade may be considered, such as formal contraindications to immunotherapy, progression on PD-1 inhibitors, or solid-organ transplant recipients. This addition addresses the concern about a “singular treatment story” while preserving the immunotherapy-focused scope of the review.

Reviewer 3 Report

Comments and Suggestions for Authors

Dear Authors,

I appreciate your review, as squamous cell carcinoma becomes significantly more difficult to treat in advanced stages and any effort to improve knowledge regarding molecular profiling, role of environmental factors, immunotherapy and other innovative treatments are beneficial to patients. After evaluating your manuscript I formulated the following remarks:

The Abstract includes the definition of advanced cutaneous squamous cell carcinoma (aCSCC) and indicates the aspects that are reviewed, from epidemiology to new therapeutic options such as PD-1 blockade and cemiplimab, alongside with discussing important prognostic determinants.  The Introduction is clear, concise and presents in a well-structured sequence important data regarding the growing incidence, clinical spectrum, improved diagnostic methods, available treatment options, emphasizing the importance of immune checkpoint inhibitors, and evolution of this type of skin cancer. Being based on evidence from clinical trials, this narrative review is a useful synthesis for practitioners involved in the multidisciplinary management of this patients.

In Materials and Methods the literature search is explained, with an effort to include all relevant data available on databases, explaining also the exclusion criteria; the selection of keywords is argued in order to avoid restrictions related to language or type of publication, from conference abstracts to clinical trial registries.

The Results are structured in 5 sections, according to the aspects evaluated by the authors, as follows: 1. Epidemiology: gives up-to-date information on the wide-world incidence, frequency of metastasis and survival outcomes, underlining the need for a uniform definition for aCSCC in order to have objective evaluations. 2. Pathogenesis and Evolution to Advanced cutaneous squamous cell carcinoma (cSCC): presents the initiating factors responsible for carcinogenesis, such as environmental conditions, genomic alterations and impaired immune response. The authors describe the factors that play a central role in the invasion and dissemination mechanisms, from functional impairment to gene mutations that sustain the invasive potential. 3. Clinical Features, Diagnostic Work-up, Histology, and Staging: gives the definition of aCSCC and presents the anatomical sites with high-risk and the factors influencing the invasion, regional spread, staging and treatment options. The scientific value of this narrative review is endorsed by the introduction of figures and tables summarizing the most important risk factors and parameters associated with aggressive lesions. 4. Therapeutic Strategies in Advanced cSCC : the authors highlight the role of a multidisciplinary approach based on current guidelines, which are still needing more randomized evidence. They also present the role of surgery, radiotherapy, non-immunotherapy alongside with translational and emerging therapies. 5. Immune Checkpoint Inhibitors in Advanced cSCC: this section is dedicated to new treatment options that revolutionized the management of this type of skin cancer, focusing on the use of cemiplimab as PD-1 inhibitor. The authors describe the mechanisms of PD-1inhibition, summarize worldwide available clinical trials and evaluate the safety considerations with a focus on immune related adverse events. A synthesis of the clinical trials exploring cemiplimab in advanced cSCC is presented in a separate table, enhancing quick evaluation and understanding by showing the design, type of disease, treatment outcome and survival rate.

In Discussion the authors start with an analysis of healthcare costs of aCSCC evolution over the last decade and underline the importance of using a therapeutic agent such as cemiplimab that can improve the prognosis of this type of cancer. However, the integration of this new therapy in the complex management of these patients is not well documented yet and is a subject of future clinical trials. However, this section is very long and it would be better structured in separate paragraphs, in which it would be easy to identify the main advancements related to different aspects important for clinicians, such as clinic-demographic variables, disease extend at baseline, better histopathologic stratification or life-style related variables.

The Conclusions are too general and based on the evaluation of many clinical trials one would expect some more precise statements and recommendations.

Author Response

We sincerely thank the Reviewer for the thorough and thoughtful evaluation of our manuscript, and we are very grateful for the extremely positive feedback and for appreciating the value of our work. In accordance with the Reviewer’s insightful suggestions, we have revised and refined the Conclusions to better reflect the key messages and clinical implications of the review.

Round 2

Reviewer 2 Report

Comments and Suggestions for Authors

The manuscript can be accepted